# Allometric Growth of Annual *Pinus yunnanensis* After Decapitation Under Different Shading Levels

**DOI:** 10.3390/plants14152251

**Published:** 2025-07-22

**Authors:** Pengrui Wang, Chiyu Zhou, Boning Yang, Jiangfei Li, Yulan Xu, Nianhui Cai

**Affiliations:** 1The Key Laboratory of Forest Resources Conservation and Utilization in the Southwest Mountains of China Ministry of Education, Southwest Forestry University, Kunming 650233, China; 18387876686@163.com (P.W.); zhouchiyu@swfu.edu.cn (C.Z.); yangboning@swfu.edu.cn (B.Y.); ljfei9661@126.com (J.L.); xuyulan@swfu.edu.cn (Y.X.); 2Key Laboratory of National Forestry and Grassland Administration on Biodiversity Conservation in Southwest China, Kunming 650224, China

**Keywords:** *Pinus yunnanensis*, photosynthesis, genetic factors, growth characteristics

## Abstract

*Pinus yunnanensis*, a native tree species in southwest China, is shading-tolerant and ecologically significant. Light has a critical impact on plant physiology, and decapitation improves canopy light penetration and utilization efficiency. The study of allometric relationships is well-known in forestry, forest ecology, and related fields. Under control (full daylight exposure, 0% shading), L1 (partial shading, 25% shading), L2 (medium shading, 50% shading), and L3 (serious shading, 75% shading) levels, this study used the decapitation method. The results confirmed the effectiveness of decapitation in annual *P. yunnanensis* and showed that the main stem maintained isometric growth in all shading treatments, accounting for 26.8% of the individual plant biomass, and exhibited dominance in biomass allocation and high shading sensitivity. These results also showed that lateral roots exhibited a substantial biomass proportion of 12.8% and maintained more than 0.5 of higher plasticity indices across most treatments. Moreover, the lateral root exhibited both the lowest slope in 0.5817 and the highest significance (*p* = 0.023), transitioning from isometric to allometric growth under L1 shading treatment. Importantly, there was a positive correlation between the biomass allocation of an individual plant and that of all components of annual *P. yunnanensis*. In addition, the synchronized allocation between main roots and lateral branches, as well as between main stems and lateral roots, suggested functional integration between corresponding belowground and aboveground structures to maintain balanced resource acquisition and architectural stability. At the same time, it has been proved that the growth of lateral roots can be accelerated through decapitation. Important scientific implications for annual *P. yunnanensis* management were derived from these shading experiments on allometric growth.

## 1. Introduction

Biomass represents a core biochemical and mechanistic attribute of vegetation, and it reflects the status of resource allocation and the ability to use environmental resources [1]. Biomass allocation is the necessary driving force for the combined consequences of carbon assimilation efficiency [2]. At the genetic, physiological, functional, and developmental levels, allometry was derived from the observation of numerous similar traits in organisms, morphology, physiology, and ecology [3]. In addition, allometric growth revealed an intrinsic, scale-independent relationship in the biomass allocation characteristics of plant components [4]. Moreover, the allometric relationship is well-known in forestry, forest ecology, and related fields [5]. A number of researchers have conducted studies on the allometric growth of *Fraxinus mandshurica* seedlings from different provenances [6], as well as on the forest biomass allocation in China based on the allometric database [7]. The allometric growth of *Salix gordejevii* branches in the sandy habitat of northern China has been studied [8]. In addition, although the effect of nitrogen and phosphorus compound fertilizers on biomass allocation and allometric growth of *P. yunnanensis* after decapitation has been studied [5], researchers still need to further investigate the allometric growth patterns of *P. yunnanensis* across different shading levels. At the same time, under different shading treatments, the allometric growth of *P. yunnanensis* after decapitation still needs to be studied. However, allometric theory fundamentally describes how plants’ shape and size scale predictably during ontogeny [5]. The relative growth relationships among organs can be obscured when biomass allocation ratios differ markedly [9]. Biomass partitioning analysis by itself might miss key aspects of plant resource economics [9]. Therefore, studying the relationship between biomass and allometric growth of *P. yunnanensis* can clarify the growth characteristics of plants.

In the natural environment, the development of plants is strongly influenced by light as a critical ecological parameter [10]. When exposed to shade, most plants undergo modifications in certain morphological and physiological traits, a process termed the shade-avoidance response [11]. Carbon accumulation and distribution patterns are altered when shading modifies leaf photosynthesis, thereby affecting plant nutrient requirements [12]. Studies showed that moderate shading boosts seedling growth, increases photosynthetic efficiency, and promotes essential nutrient accumulation [13]. However, the decapitation of individual *P. yunnanensis* plants effectively stimulates basal adventitious bud growth and enables the acceleration of canopy formation through root nutrient supply [5]. In addition, decapitation improves the light levels, decreasing leaf shading and promoting the use of solar radiation for productivity [5]. Under different shading levels, a series of structural changes were rapidly induced following light detection in plants [10]. Moreover, shading affects plant biomass by regulating plant’s demand for nutrients [11]. The result of the study on *Fritillaria thunbergii* showed that bioactive compound levels were elevated by 11.5–11.9% under shaded conditions, resulting in 9.5–13.0% less biomass and reduced active components [14]. Light conditions also affect root growth and development [15]. When plants photosynthesize sufficiently, they allocate surplus photosynthates to their root systems, which stimulated root growth [15]. Under shading treatments, plants focus carbon resources on growing leaves rather than roots or stems, helping them adapt faster to low-light conditions [16]. Experimental data show that the shading treatment impaired root development in *Arabidopsis thaliana*, while its aboveground biomass accumulation and overall growth also showed parallel weakening trends [17]. Since photosynthesis predominantly occurs in leaves, plants respond to shading by increasing needle area, a strategy that promotes carbon gain and elevates photosynthetic efficiency [17]. To sustain growth in shading, plants strategically shift carbon allocation, and they decrease resource investment in roots and stems while favoring needle development to boost expansion and adaptive capacity [18]. Therefore, it is of great significance to study the effect of shading on the allometric growth of *P. yunnanensis*. Nevertheless, studies of *P. yunnanensis* have mainly focused on transcriptomic sequencing, nutrient composition, characteristics of community structure, and biomass [10]. Furthermore, recent studies have shown that there is a strong correlation between the diameter and the length of *Polygonatum multiflorum* branches under shading habitat [12]. However, the effect of shading on allometric growth and biomass allocation of different organs has not been systematically studied. In conclusion, analysis of seedling growth dynamics under shading conditions provides fundamental insights into plant adaptation strategies, with the case study of *P. yunnanensis* allometric growth offering particularly valuable evidence for understanding shading response mechanisms.

*P. yunnanensis* is a special conifer in southwest China and is famous for its high ecological environment, light preference, strong adaptability, and great natural regeneration potential [10]. It has strong adaptability and great potential for natural regeneration [10]. *P. yunnanensis* also has sexual reproduction to solve the problems of low seed setting rate and phenotypic variation [10]. At the same time, *P. yunnanensis* has been propagated through seedlings for afforestation, and sexual reproduction suffers from low seed yield, inconsistent germination rates, and variable offspring traits [11]. Asexual propagation offered a viable solution to bypass these limitations inherent in seedling-based approaches [19]. In addition, high-quality production with abundant branching provides a solid foundation for success [2]. Scientists have examined nutrient supplementation and supplemental plant growth regulators to promote superior branch development in *P. yunnanensis* [5,19]. Decapitation is the basic management of asexual reproduction in that dominance by the apical meristem is reduced, while bud activation is induced at nodal positions [5]. The system maintains an optimal supply of lateral vegetative shoots [10]. This procedure also guarantees sufficient shoot branching availability to support vegetative reproduction at the same time. However, the effects of environmental changes on biomass allocation among different organs and seedling growth in *P. yunnanensis* after decapitation remains unclear. To explore this research void, annual *P. yunnanensis* seedlings were used as materials, and the method of decapitation was adopted in this study. The biomass allocation, phenotypic plasticity, and allometric growth of the main root, lateral root, main stem, lateral branch, shoot, main stem’s needles, lateral branch’s needles, and shoot’s needles were analyzed under different shading levels. Moreover, some researchers have studied the biomass allocation and growth of other species of seedlings under different shading levels. Other species allocate more biomass to the main stem and root under different shading, and the growth of the seedling is dominated by the growth of the root [20]. However, the biomass allocation and growth of *P. yunnanensis* after decapitation under different shadings are unclear. Based on the results of biomass allocation and growth of other species under different shading conditions, we scientifically speculated that the biomass and growth status of different organs would be different under different shading conditions; among them, the lateral roots of *P. yunnanensis* may be more sensitive to light after decapitation. Meanwhile, the purpose was to study the biomass allocation and allocation of different organs and the allometric growth of individual biomass under different shading levels. The result lays a basic foundation for the natural regeneration, decapitation, biomass allocation of *P. yunnanensis* under different shading, and the restoration of the forest community.

## 2. Methodology

### 2.1. Research Site

The nursery used in this study is maintained by Southwest Forestry University located in northern Kunming, where the average annual sunshine duration is 2245.6 h and 56%. The dry season has a subhumid climate in the northern sub-tropics with monsoon on the plateau.

### 2.2. Individualized Design

The test seedlings were healthy, disease-free, and uniform annual potted seedlings of annual *P. yunnanensis*. The seedlings were decapitated, shaded, and watered regularly. An individualized experimental design was employed, comprising 4 technical replicates, each with 3 biological replicates. This design included four different shading levels, with 12 seedlings per treatment: CK (full sunlight, 0% shading), L1 (light shading, 25% shading using a two-needle net), L2 (medium shading, 50% shading with a three-needle net), and L3 (heavy shading, 75% shading with a four-needle net).

### 2.3. Collection and Treatment of Seeds

The experimental seeds were obtained from the Yuns-Cso-Py-001-2016, established in 1987 and completed in 1996. This facility, currently recognized as China’s sole annual *P*. *yunnanensis* breeding base, is geographically positioned at 100°28′ E, 25°27′ N. The site experiences an annual mean temperature of 16.2 °C with temperature extremes of 34.5 °C and −6.8 °C, maintains 70% average relative humidity, and lies at 1900–2000 m elevation. The mountainous red soil exhibits moderate fertility with a pH of 6.0. Mature current-year cones were harvested from healthy, productive parent trees. After labelling, cones were transported to the laboratory for air drying. Well-formed seeds were extracted following cone dehiscence. Seed sterilization was performed using 0.5% KMnO_4_ solution, 30 min immersion, followed by rinsing with distilled water and 24 h soaking in 50 °C warm water, and the seeds were sown and cultivated at the same time.

### 2.4. Decapitation and Shading of Seedlings

We maintained the seedlings using standard practices after germination. Subsequently, we applied varying shade levels to the cultivated seedlings: CK received full sunlight (0% shading), L1 had light shading (25%) using a two-needle net, L2 medium shading (50%) with a three-needle net, and L3 heavy shading (75%) with a four-needle net. The experimental design followed a completely randomized single-factor block arrangement.

### 2.5. Determination of Seedling Biomass

Biomass measurements of annual *P*. *yunnanensis* seedlings were conducted at 60-day intervals following stumping. From each replicate test group, three seedlings were randomly selected for each of the three sampling repetitions, totaling 8 measurement periods. The shoot branch was recorded using a ruler with a precision of 0.1 cm, while the basal stem diameter was measured with vernier calipers with the precision of 0.01 mm. During each measurement, shoot branches were excised and counted. Plant components were separated into main roots, lateral roots, main stem, lateral branches, shoot branches, main root needles, and lateral branch needles. These components were oven-dried at 80 °C until constant weight was achieved, with biomass measured to 0.001 g precision. Biomass represented the combined mass of branches and needles, while aboveground biomass included main and lateral roots, main stems, lateral branches, and needles. Belowground biomass comprised main root and lateral root masses, with total biomass being the sum of above- and belowground components. Due to the low stumping height employed in this study, which resulted in minimal needle retention, lateral branching, lateral branch and needle biomass were excluded from the analysis. For allometric modelling, shoot branch biomass served as a proxy for combined shoot branch and needle biomass.

### 2.6. Plant Growth Assessment

#### Determination of Phenotypic Plasticity Index

The experimental plants were annual container-grown *P. yunnanensis* seedlings, and 48 seedlings were taken from each treatment. We cleaned up the residual soil in all parts (main root, lateral root, main stem, lateral branch, shoot branch, main stem needle, lateral branch needle, shoot branch needle) of annual *P. yunnanensis*, the maximum and minimum values of the statistical variables, and finally the phenotypic plasticity index. The phenotypic plasticity index formula is PI = (Max − Min) ÷ Max. Moreover, max and min represent the highest and lowest mean values of a given variable.

### 2.7. Statistics and Analysis of Data

We recorded the individualized data in Excel. Secondly, we conducted one-way ANOVA and Pearson correlation analysis on the biomass of different organs and individual plants of annual *P. yunnanensis* by using IBM SPSS Statistics 27 software. Duncan’s new multiple-range test was used to study the changes of biomass of different organs and individual plants of annual *P. yunnanensis* under different shading levels. We used SMART 2024 software to analyze the biomass of each organ under different levels and the linear relationship between individual plant biomass and allometric growth. The coefficient of variation was expressed as standard (error-mean) × 100% for data plotting. Finally, we used the Origin 2025 software to analyze the coupling relationship between four technical replicates and three biological replicates, and the biomass of different organs under different levels.

## 3. Results

### 3.1. Effects of Shading on Biomass Allocation

Under different shading levels, with the increase in shading levels, the allocation of biomass was significantly affected, and the significance decreased gradually. Among all treatments, the CK treatment exhibited the highest biomass accumulation, with the most statistically significant differences (*p* < 0.01) (Figure 1), and the proportion of biomass of different organs in the individual plant biomass was also affected (Figure 2). The main stem biomass accounted for the largest proportion of the individual plant biomass, followed by the needle biomass. Moreover, the shoot branches exhibited a substantial biomass proportion of 12.2%, comparable to that of lateral branch needles, indicating vigorous growth activity of newly formed shoots following decapitation. Overall, the plant allocated the majority of its biomass to aboveground woody organs, while photosynthetic needles accounted for a relatively smaller proportion of 18.4%. In addition, with the increase in shading intensity, the biomass ratio of the main stem and main stem’s needle decreased first and then increased, while the biomass ratio of lateral branches and lateral roots increased first and then decreased. In contrast, for the main stem and lateral root biomass, the proportion of main stem biomass to individual plant biomass was the largest, but the proportion of lateral root biomass to individual plant biomass was the smallest under L3 shading treatment; under L2 shading treatment, the ratio of main stem biomass to individual plant biomass was smaller, and the ratio of lateral root biomass to individual plant biomass was the largest. It could be inferred that there may be a negative correlation between the biomass allocation of main stem and lateral roots. Meanwhile, *P. yunnanensis* will adjust the proportion of biomass in different organs to adapt to different shading treatments (Figure 3). This allocation pattern likely reflects adaptive resource partitioning strategies in response to the decapitation treatment. Therefore, we speculated that to outcompete other plants for sunlight, annual *P. yunnanensis* might strategically direct more biomass towards its main stem and needles, thereby increasing the height of the main stem and minimizing the risk of being overshadowed by neighboring plants.

### 3.2. Relationship Between Phenotypic Plasticity Analysis and Biomass Allocation Under Different Shading Levels

As shown in Table 1, distinct patterns in phenotypic plasticity were observed across different organs of annual *P. yunnanensis* under varying shading treatments. For instance, lateral branches maintained higher plasticity indices across most treatments, while the main stem displayed consistently lower plasticity, particularly under L1 shading, where its plasticity index dropped below 0.5, reflecting constrained vertical growth under light limitation. In contrast, lateral branches demonstrated greater responsiveness, maintaining higher plasticity indices across most treatments, though similarly reduced below 0.5 under L1 treatments. Needle biomass exhibited intermediate plasticity levels, generally exceeding main stem values but remaining below those of lateral branches. These different reflections highlighted how *P. yunnanensis* seedlings preferentially adjust lateral branch and needle biomass allocation rather than modifying main stem growth when adapting to shading gradients. The notable reduction in plasticity across all organs under L1 treatment suggests a threshold response to initial shading exposure, while the subsequent recovery under heavier shading treatment shows complex acclimation dynamics in this species.

### 3.3. Allometric Relationships Between Different Components and Plant Biomass Under Different Shading Levels

There was a close relationship between biomass allocation and allometric growth. Table 2 illustrates that lateral roots exhibited allometric growth under light shading (L1), characterized by a lower slope of 0.5817 and higher significance (*p* = 0.023), suggesting this light shading intensity most strongly constrained lateral root allocation. Under other shading levels, lateral roots showed different growth patterns, transitioning from isometric to allometric growth. Furthermore, the intercepts from 0.8406 to 0.9736 indicate baseline size differences among treatments. In summary, these patterns collectively imply that shading alters root allocation strategies in annual *P. yunnanensis*, with effects being nonlinear across light gradients.

Table 3 shows that the main stem’s growth rate was isokinetic under different shading conditions. While all treatments showed isometric growth, their regression characteristics varied markedly. The L2 and L3 treatments demonstrated exceptionally strong correlations with near-isometric slopes from 1.0276 to 1.1722, and indicated proportional main stem growth relative to the reference variable under heavier shading. In contrast, the CK and L1 treatments exhibited weaker fits and negative allometry, suggesting light-limited conditions promote relatively reduced stem allocation. These results collectively suggest that while main stems generally maintain isometric scaling, shading intensity non-linearly modulates both the strength of allometric relationships and absolute growth patterns in annual *P. yunnanensis*. They also indicate that under different shading levels, annual *P. yunnanensis* when decapitated mainly focuses on the allocation of biomass to the main stem, thereby affecting the partitioning of biomass to lateral roots. The allometric growth of lateral roots under different shading levels showed different growth effects.

Figure 4 shows that the growth patterns of annual *P. yunnanensis* roots and stems under shading gradients revealed organ-specific strategies. Main and lateral roots exhibited light-dependent plasticity, transitioning from isometric to allometric growth under moderate and light shading, respectively. This shift likely reflects prioritized resource allocation to root systems for enhanced nutrient uptake as light becomes limiting. In contrast, the main stem maintained strict isometric growth across all treatments, and its role as a structural priority was unaffected by shading. Meanwhile, lateral branches grew faster as shading increased but maintained proportional scaling isometry. This pattern suggests they expanded horizontally to capture more light under lower irradiance conditions. These coordinated yet distinct responses highlight how belowground and aboveground structural organs differentially adapt to light constraints.

Figure 5 mainly indicates that needle biomass showed varying responses to shading intensity. Main stem needles switched from allometric to isometric growth as shading increased. Lateral branch needles showed allometric growth only in moderate shade, optimizing light capture. These differential needle responses between stem and branch positions demonstrate how annual *P. yunnanensis* fine-tunes its photosynthetic architecture across microenvironments created by shading.

Figure 6 illustrates that there was a positive correlation between the biomass allocation of individual plants and that of all components of annual *P. yunnanensis*. While the positive correlation between individual plant biomass and component allocations reflects fundamental coordination in growth regulation, the treatment-specific variations revealed more nuanced adaptive responses. The synchronized allocation between main roots and lateral branches, as well as between main stems and lateral roots, suggests functional integration between corresponding belowground and aboveground structures to maintain balanced resource acquisition and architectural stability. Particularly noteworthy is the negative relationship under L2 shading, where reduced investment in shoot branch needles indicates a shift in allocation priorities when light becomes limiting. This pattern implies threshold-dependent reallocation from photosynthetic surfaces to structural components, likely to optimize light interception efficiency under moderate shading conditions. These findings collectively highlight how annual *P. yunnanensis* dynamically adjusts its biomass partitioning between structural growth and photosynthetic capacity across light gradients, with shading intensity serving as a key modulator of these source–sink relationships.

## 4. Discussion

### 4.1. Factors Affecting Biomass Allocation

Biomass allocation in plants is influenced by a combination of environmental and genetic factors. In China, numerous studies have shown that understory plants tended to allocate more biomass belowground under specific climatic conditions [21]. China is categorized into five distinct climatic zones, among which there are three in southwest China: zones with distinct hot summers and cold winters, cold zones, and temperate zones [22]. The climate sensitivity and ecological vulnerability of southwest China have the attributes of climate change and are highly vulnerable to recurrent extreme climate levels [23], and the results indicated that cumulative drought inhibits and promotes 57.3% and 25.0% of vegetation productivity with significant differences among different environments and among different vegetation types [24]. Additionally, studies on *Gentianella turkestanorum* have demonstrated that increasing altitude leads to decreased temperatures, slower growth rates, and reduced respiratory metabolic activity, ultimately resulting in lower biomass at higher altitudes [25]. Therefore, climate is one of the factors that affects biomass allocation. According to the investigation of relevant departments, the average temperature of the dry season in Kunming is 5.5–18.3 °C, and the average humidity is recorded at 66%, with rainfall averaging 21.4 mm; the average temperature of the rainy season is 15.1–23.3 °C [26]. Climate data reveal 79% relative humidity and 147.1 mm rainfall on average, comprising 85% of annual totals [27]. Therefore, this study was conducted under the subtropical upland climate of Kunming, where sufficient photosynthetically active radiation during the annual sunshine period contribute to increasing total plant biomass and positive carbon balance, even under experimental shading conditions. Moreover, the mild annual mean temperatures and distinct wet and dry seasons provide optimal conditions for biomass allocation between structural and photosynthetic tissues. Collectively, these environmental factors explain the observed allometric patterns, where light limitation below 50% shading triggered a significant redistribution of biomass from stems to needles, resulting in a significant increase in the number of shoot-to-shoot seedlings. This phenomenon highlights the species’ phenotypic plasticity in resource optimization.

In addition, allometric biomass distribution is determined by both environmental situations and genetic influences [28]. Previous research showed that a clear evolutionary strategy shaped Turkey grass to invest 59.24% of its biomass in flower production [17]. In this study, among the four different shading levels, the main stem biomass constituted the highest proportion of the total individual plant biomass. Consequently, biomass allocations are different [29]. However, extreme conditions trigger plants to shift resources toward main stem needles, favoring survival through improved photosynthesis at the expense of propagation and growth [30]. The results suggest that *P. yunnanensis* might preferentially allocate biomass to the main stem and needles to enhance light competition, promoting vertical growth to avoid shading by neighboring plants in this study. The biomass allocation patterns of *P. yunnanensis* showed different organ-specific allocations, and this hierarchical allocation strategy reflects three key adaptation mechanisms, a structural investment mechanism for disproportionate main stem allocation, and a structural investment mechanism for main stem allocation, thus enhancing vertical growth in light competition, consistent with the conifer carbon optimization hypothesis; in addition, conifer biomass maintained more than 20% allocation even under 75% shading, suggesting that the growth of these conifers is more efficient than that of other conifers, suggesting photosynthetic optimization mechanisms for light adaptation through needles area expansion; and finally, a strong negative correlation between main stem and lateral root biomass demonstrates the role of functional balance theory, which could be used as a reference for future research; the limited carbon was strategically allocated to the organs most critical to current light conditions. Together, these patterns illustrate the evolutionary adaptation of this species to a forest belowground environment in which simultaneous investment in high growth and daylighting structures maximizes fitness, providing a model for the adaptation of this species to the forest belowground environment, especially under competitive shade conditions.

Moreover, the presence of shading may lead to shifts in nutrient allocation and stoichiometric traits and affect biomass allocation [31]. Plants rely on carbon (C), nitrogen (N), and phosphorus (P) for growth, and the way these nutrients are partitioned among organs reveals their adaptive tactics and physiological reactions to environmental factors [32]. Biomass allocation is affected by photosynthesis, which depends on proteins and enzymes synthesized using nitrogen and phosphorus [33]. In this study, it was found that the main stem biomass accounted for the largest proportion of plant individual biomass, followed by needle biomass. In addition, the biomass ratio of aboveground branches was 12.2%, which was equivalent to that of lateral branches and needles. Therefore, we can speculate that *P. yunnanensis* might dynamically adjust the main stem and needles’ storage content of C, N, and P in different organs to adapt to the effects of different shading treatments on photosynthesis and then regulate the allocation of biomass. In summary, annual *P. yunnanensis* responds to environmental variability by dynamically regulating the storage and distribution of C, N, and P among different organs, with these stoichiometric changes necessarily resulting in biomass allocation shifts.

### 4.2. The Relationship Between Phenotypic Plasticity Analysis and Plant Allometric Growth

The concept of phenotypic plasticity has been widely utilized by researchers to elucidate the universal principles governing the adaptability of living organisms [34]. In addition, phenotypic plasticity can produce large geographic and seasonal differences among and across populations [35]. Such phenotypic variation may be determined by the concentration and characteristics of the entities involved in the ecological network and modify the results of interdependent evolution [36]. Researchers have discussed the use of phenotypic plasticity analysis in natural vertebrate communities, which also showed how evolutionary pressures and environmental constraints can change the selection and adaptation of a population to the environment through their fitness consequences [25]. Furthermore, research on carotenoids has shown that phenotypic plasticity changes the distribution of phenotypes, thus affecting the direction and intensity of selection [37]. In this plasticity study, the lateral branches showed greater responsiveness and maintained a high phenotypic plasticity index under most treatments, and the coniferous biomass showed a moderate level of plasticity. Therefore, we can speculate that after decapitation, the biomass of *P. yunnanensis* changes to adapt to different shading treatments, resulting in a significant increase in the phenotypic plasticity index of lateral branches and needles. Therefore, the growth of lateral branches is promoted, and the adaptability is improved, which affects the change in *P. yunnanensis* population. Therefore, it is of great significance to study the phenotypic plasticity analysis of the biomass of each organ of *P. yunnanensis* after decapitation. After conducting a comprehensive analysis of this study, a strong correlation is found between population evolution and adaptability, as evidenced by lateral branches and needles.

All the biomass distribution characteristics varied with organ size, and most of them showed significant interactions between organs of different sizes and light [38]. With the increase in organ size, the difference between different light levels also increased [26]. Studies have shown that species exhibit a variety of biomass allocation patterns under different light regimes, and these results underscore the importance of morphological traits in variation [39]. The results of predawn biomass allocation of backcrossed *Castanea americana* seedlings under different light intensities showed that partial shading could assist American chestnut in enduring drought at the early establishment stage by influencing physiological regulation [40]. Additionally, in a study on shade-tolerant tree seedlings grown in tropical forests, the results showed that with increasing shading and branch allocation, the number of leaves allocated was much lower than that observed here [41]. Both low and excessive light will affect the physiological characteristics of plants [42]. Other studies have shown that needle mass, which served as a determinant for biomass allocation, is generally higher in low-light environments. In contrast, biomass allocation patterns among different organs are modified through functional and structural changes in full-light conditions. This allocation strategy enables them to make better use of bright light resources and withstand the environmental pressures associated with increased light exposure [43]. Based on the analysis of four shading gradients in this study, we found that lateral branches exhibited accelerated growth rates under increasing shade treatments while maintaining consistent size scaling patterns. This morphological response suggests that the observed horizontal expansion of lateral branches likely represents an adaptive strategy to enhance light capture under reduced radiation conditions, thereby promoting greater biomass allocation to branch structures. The differential growth responses between aboveground lateral branch elongation and belowground organs underscore their distinct acclimation strategies to light limitation. Specifically, the shift in biomass partitioning toward lateral branches under shading treatments reflects a functional adaptation to optimize light interception while maintaining structural balance within the individual plant system. These findings align with the broader pattern observed across species where plants modify organ-specific biomass allocation to cope with varying light environments yet unique phenotypic plasticity in annual *P. yunnanensis* was demonstrated through its isometric maintenance of branch dimensions despite accelerated growth rates.

The concept of correlation in plants refers to the dynamic balance between suppression and synergy in the growth of different organs [44]. Optimal growth is achieved through environmentally mediated biomass allocation, where plants invest more in organs critical for overcoming resource deficits [10]. Each plant part has unique traits shaped by its growth stage and environment [10]. In the present study, we found a positive correlation between the biomass allocation of individual plants and the biomass allocation of all components of *P. yunnanensis*, which reflects the basic coordination of growth regulation, and the relationship between the biomass allocation of individual plants and the biomass allocation of all components of *P. yunnanensis*, but the variation revealed a more subtle adaptive response. In addition, we can also speculate that there is a functional integration between belowground and aboveground structures through synchronous partitioning between main roots and lateral branches and between main stems and lateral branches and that there is a functional integration between the aboveground and underground structures through synchronous partitioning between main roots and lateral roots, thus maintaining balanced access to resources and building stability. In summary, our results show that different shading intensities significantly changed the coordination of allometric growth among organs of annual *P. yunnanensis*, and the modified allometric growth pattern under shading conditions has a direct impact on breeding programs. The maintenance of stem length under different treatments was observed to indicate the genetic stability of apical dominance; in addition, enhanced lateral root plasticity under 25–50% shading conditions determined the optimal light conditions for root development. These findings enable precise optimization of nursery light conditions to simultaneously maximize biomass production and improve stem shape quality.

### 4.3. How Shading Intensity Affects Plant Allometric Growth

Photosynthesis is powered by light, which is essential for plants, with growth, biomass accumulation, and nutrient distribution being significantly influenced by this energy source [45]. As highlighted in the Introduction, light drives photosynthetic allocation; our data further reveal that shading intensity non-linearly modulates this process in *P. yunnanensis*. The relationship between shading intensity and plant growth was demonstrated through observable modifications in plant phenotype across multiple organizational levels [46]. Diminished light availability may curtail photosynthesis, limiting energy provision for shoot development and consequently restraining branch growth [47]. Research has demonstrated that light intensity significantly affects shoot branch growth as a key environmental determinant [47]. Researchers found that shading treatment inhibits both cytokinin biosynthesis and carbohydrate storage in sorghum, consequently maintaining bud dormancy and preventing branch outgrowth [48]. Our investigation revealed distinct patterns in how *P. yunnanensis* responds to varying shade conditions. As shading intensity increased, we observed progressively stronger effects on biomass partitioning among different plant organs, though these impacts displayed diminishing marginal significance at higher shade levels. This nonlinear response suggests the species employs compensatory mechanisms to mitigate severe light limitation. Crucially, the reduced light availability not only altered biomass distribution but also substantially decreased carbohydrate accumulation and bioenergy storage across all examined tissues. These metabolic constraints directly impaired the plant’s capacity to maintain optimal biomass allocation patterns, ultimately restricting overall growth performance. The suppression of energy reserves and structural growth highlight light’s role as a dual regulator of carbon economy and development in this conifer species.

Additionally, shading may alter the root architecture of *P. yunnanensis*. The way plants optimize nutrient uptake is mirrored in their root structural adaptations and growth patterns [49]. Light availability differentially modifies the expression of functional root system features [50]. Under low-light stress, root growth and development are often severely impaired as plants allocate more resources to aboveground expansion for improved photosynthesis [51]. Moreover, light deprivation leads to carbon partitioning that favors shoots over roots, thereby diminishing the root-to-shoot biomass ratio and compromising root function and structure [52]. Our study demonstrated that shading treatments induce significant architectural modifications in decapitated *P. yunnanensis*, particularly through contrasting responses between aboveground and belowground organs. The observed high growth rate and plasticity index of lateral branches, in contrast to the suppressed growth and low plasticity of main roots, reveal a strategic resource reallocation under light limitation. This different response suggests that while primary root development is constrained, plants compensate by enhancing lateral root proliferation, an adaptive response that expands the root surface area to maintain nutrient acquisition efficiency despite reduced photosynthetic input. Such morphological adjustments align with the carbon partition paradigm, where light-deprived plants prioritize shoot expansion over primary root growth while still optimizing absorptive capacity through finer root branching. These coordinated yet distinct growth responses between shoot and root systems illustrate how *P. yunnanensis* achieves functional balance under shading stress through modular plasticity.

The allocation of plant biomass and their effects on tree allometric growth depend mainly on access to light [53]. In agroforestry systems, allometric growth of shade tree species has been studied [52]. The results show the allometric growth of shade tree species is associated with increased light availability. In conclusion, light promotes crown expansion and corresponding stem thickening to provide mechanical support [43]. Nevertheless, shading intensity is more strongly associated with stand abundance and species count rather than diversity indices, implying that the stem density of individual organisms can be affected by shading [52]. Hence, the shading condition affects the allometric growth of trees by affecting the structural characteristics of trees [53]. Our study elucidated how shading intensities modulate allometric growth and biomass allocation in decapitated annual *P. yunnanensis* seedlings through distinct physiological mechanisms. Quantitative analyses revealed that moderate shading induced substantial morphological reprogramming, resulting in a 2.3-fold increase in lateral branch production and reflecting enhanced branching plasticity under light-limited conditions. The disruption of apical dominance initiates source-sink reallocation, preferentially directing light assimilation toward lateral organs. Concurrently, irradiance-dependent carbon partitioning fine-tunes resource distribution, particularly under low-light treatments. These adaptive responses collectively enable *P. yunnanensis* to optimize architectural design for improved light interception while maintaining structural stability. The observed patterns align with established principles of shading adaptation in woody plants, where light availability primarily governs crown expansion and associated stem development. Importantly, shading-induced modifications predominantly influence organ-specific scaling relationships rather than overall stand diversity, highlighting the species’ capacity for phenotypic integration in response to light gradients. These findings provide a physiological framework for optimizing nursery shading regimes and inform breeding strategies targeting this ecologically important conifer’s shading-adaptive traits.

## 5. Conclusions

Across a shading gradient (0%, 25%, 50%, 75%), different organs of annual *P. yunnanensis* had different sensitivities to light. Therefore, the biomass allocation of different organs was affected by shading, which changed the biomass of individual plants, resulting in an isokinetic (*p* ≥ 0.05), allometric (*p* < 0.05) plant growth type, or an allometric (*p* < 0.05) plant growth type. Additionally, shading had the greatest impact on the growth of the main stem and lateral roots. The growth rate of the main stem increased with the increase in shading, and its growth type was isokinetic growth (*p* > 0.05). Lateral roots exhibited significantly faster growth compared to the main stem, and the growth pattern was close to allometric (*p* < 0.05). Therefore, the biomass of individual plants increased with different shading levels, resulting in increased biomass across organs, with the main stem contributing the highest proportion to total plant biomass. The growth rate of lateral roots was significantly different from that of individual plants, and the growth type was allometric growth under light shading (L1). According to the results of the present study, annual *P. yunnanensis* belongs to a shade-tolerant tree species. According to related data analysis, annual *P. yunnanensis* has a strong growth and branching ability. In addition, under medium shading (L2), annual *P. yunnanensis* exhibited strong growth and branching ability, with ideal biomass allocation and distribution among each organ, which was conducive to plant growth. It has been proved that the removal of apical dominance is beneficial to the growth of lateral roots. In conclusion, this study on the allometric growth of annual *P. yunnanensis* across varying shade levels supplies a theoretical framework for the nursery management of annual *P. yunnanensis*.

## Figures and Tables

**Figure 1 plants-14-02251-f001:**
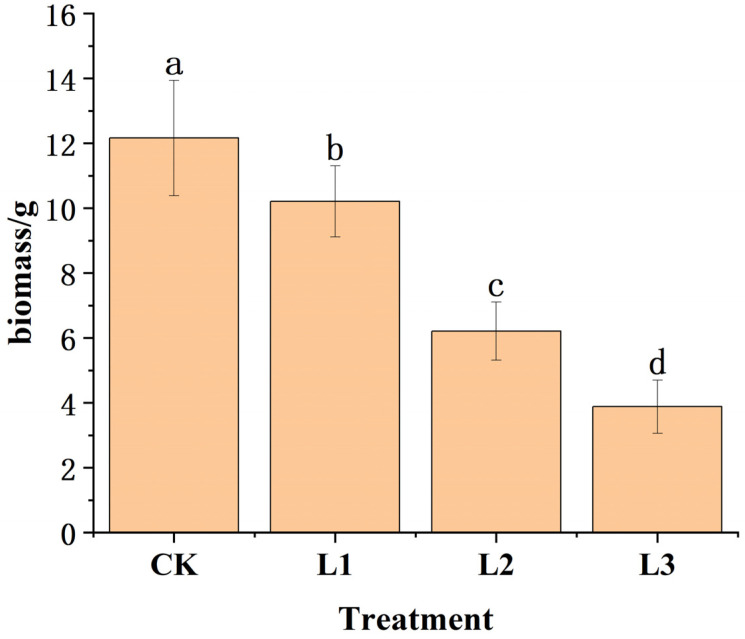
Changes in biomass of *P. yunnanensis* with different shading levels after decapitation (note: the order of lowercase letters indicates the significance of plant biomass under different treatments).

**Figure 2 plants-14-02251-f002:**
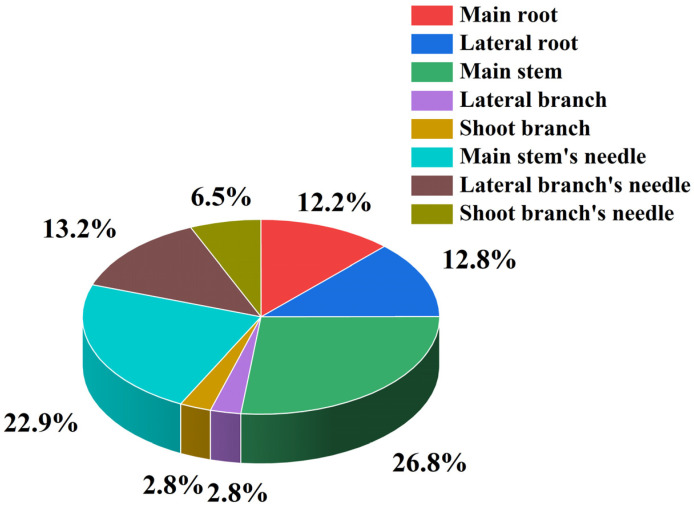
The biomass ratios of different organs to individual plants of *P. yunnanensis* after decapitation.

**Figure 3 plants-14-02251-f003:**
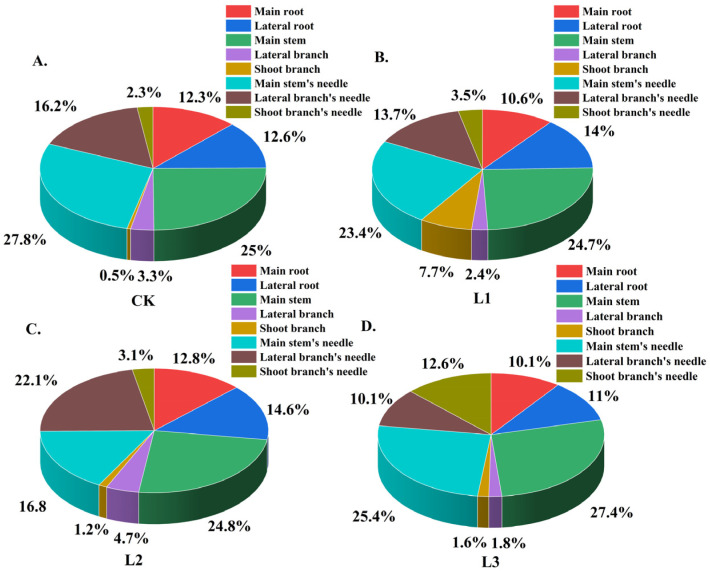
Biomass ratio of different organs to an individual plant of *P. yunnanensis* after decapitation under different shading levels. (**A**) CK received solar radiation (0% shading); (**B**) L1 had light shading (25%) using a two-needle net; (**C**) L2 medium shading (50%) with a three-needle net; (**D**) L3 heavy shading (75%) with a four-needle net.

**Figure 4 plants-14-02251-f004:**
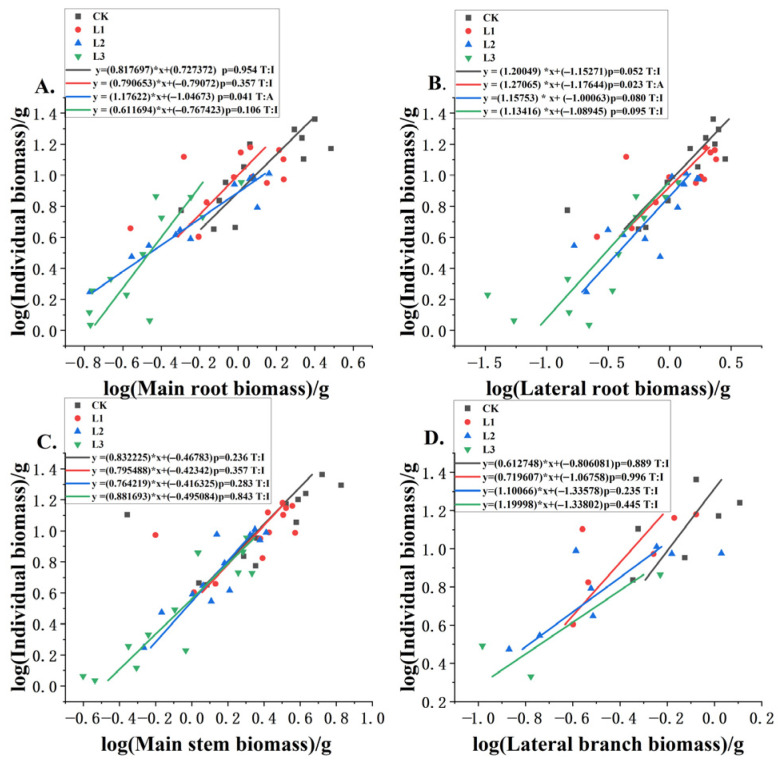
Biomass allocation patterns were assessed for (**A**) main roots, (**B**) lateral roots, (**C**) main stems, and (**D**) lateral branches relative to total plant biomass at different shade levels. Slope deviations from isometry (1.0) were tested for significance (*p*), and relationships were categorized (T) as allometric (A) or isometric (I).

**Figure 5 plants-14-02251-f005:**
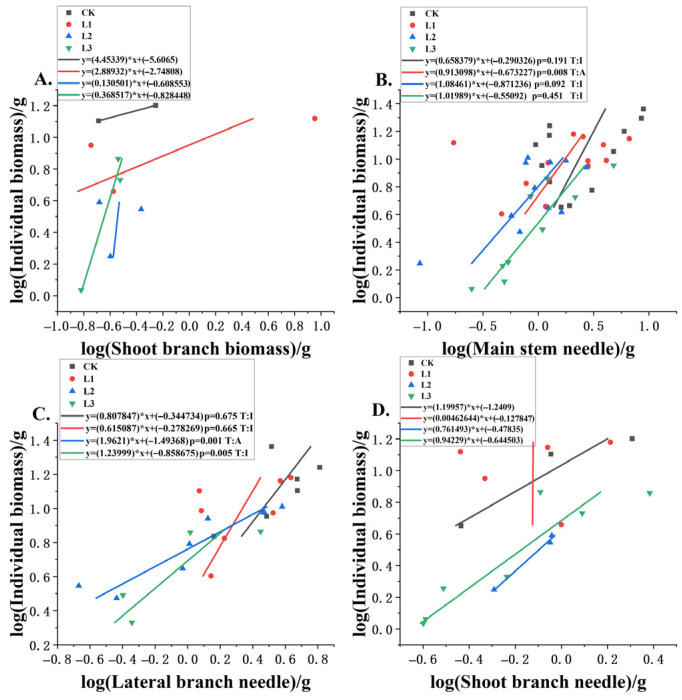
The allometric relationships between the biomass of an individual plant and the biomass of (**A**) shoot branches, (**B**) main stem needles, (**C**) lateral branch needles, and (**D**) shoot branch needles under different shading levels.

**Figure 6 plants-14-02251-f006:**
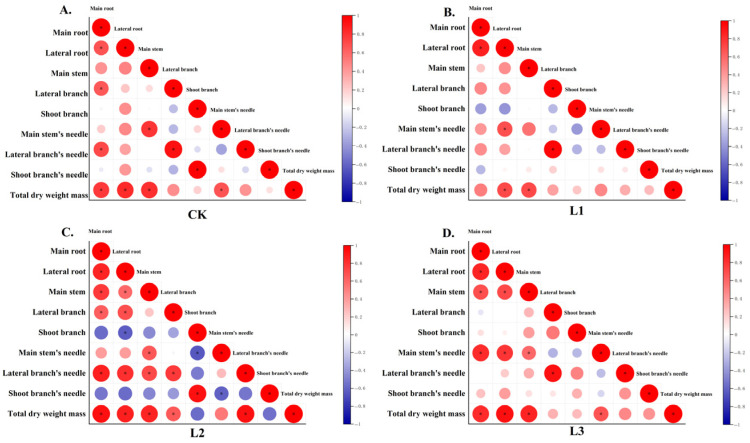
Correlation of biomass allocation among components of *P. yunnanensis* across varying shade levels. (**A**) CK received solar radiation (0% shading); (**B**) L1 had light shading (25%) using a two-needle net; (**C**) L2 medium shading (50%) with a three-needle net; (**D**) L3 heavy shading (75%) with a four-needle net. The significant difference between the two is indicated by *.

**Table 1 plants-14-02251-t001:** The phenotypic plasticity index of different organs under different shading levels.

PI	Main Root	Lateral Root	Main Stem	Lateral Branch	Shoot Branch	Main Stem Needle	Lateral Branch Needle	Shoot Branch Needle	Individual Biomass
CK	0.58	0.65	0.77	1.00	1.00	0.85	1.00	1.00	0.63
L1	0.69	0.73	0.30	0.78	1.00	0.86	0.83	0.71	0.33
L2	0.63	0.62	0.58	0.83	1.00	0.81	0.84	1.00	0.60
L3	0.74	0.76	0.79	1.00	1.00	0.85	1.00	1.00	0.78

**Table 2 plants-14-02251-t002:** Allometric relationship of lateral roots under different shading levels.

Lateral Root	n	R^2^	P	Slope	Intercepts	H0_b	F	*p*	Type
CK	12	0.634	0.002	0.6632	0.9736	1.000	4.871	0.052	I
L1	12	0.546	0.006	0.5817	0.9386	1.000	7.129	0.023	A
L2	12	0.669	0.001	0.7064	0.8406	1.000	3.794	0.080	I
L3	12	0.661	0.001	0.7168	0.8692	1.000	3.391	0.095	I

**Table 3 plants-14-02251-t003:** Allometric relationship of the main stem under different shading levels.

Main Stem	n	R^2^	P	Slope	Intercepts	H0_b	F	*p*	Type
CK	12	0.372	0.035	0.7329	0.7431	1.000	1.588	0.236	I
L1	12	0.395	0.029	0.7903	0.6966	1.000	0.933	0.357	I
L2	12	0.803	0.000	1.1722	0.5645	1.000	1.289	0.283	I
L3	12	0.821	0.000	1.0276	0.5531	1.000	0.042	0.843	I

## Data Availability

All data generated or analyzed during this study are included in this article. All data generated or analyzed during this study are available from the first author or request.

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
