# Peer review of "Allometric Growth of Annual *Pinus yunnanensis* After Decapitation Under Different Shading Levels"

_plants, 2025, doi:10.3390/plants14152251_

Round 1
Reviewer 1 Report
Comments and Suggestions for Authors
Comments to author
In the abstract section, the start part of the abstract is review data, the middle part result,,, please increase middle part add more result, reduce start section
line 144:2.4. decapitation and shading the seedlings, d capital
the manuscript seems interesting and is written perfectly.
Reviewer 2 Report
Comments and Suggestions for Authors
Review of the article by Peng Rui Wang et al.: "Allometric growth of Pinus yunnanensis after decapitation under different shading levels"
The paper examines the effect of decapitation on allometric relationships in Pinus yunnanensis under different light growing conditions. A detailed assessment of the distribution of biomass by plant organs has been carried out. It was shown that the biomass of the main stem accounted for the largest share in the biomass of an individual plant and had the main influence on the distribution of P. yunnanensis biomass. The levels of shading affected the distribution of biomass in the main stem and the growth of the lateral roots. Experiments on allometric growth with shading allowed us to draw important conclusions for the control of P. yunnanensis. At the same time, I would like to see in the article the average values of full sunlight in the growing region. In addition, do the authors have data on the distribution of biomass among plant organs under different light conditions? This would be interesting in terms of evaluating the donor-acceptor relationships in the system of the whole plant. (In addition to Figure two). The authors write The phenotypic plasticity index formula is PI= (Max-Min) ÷ Max. It makes sense to give this formula in the form PI= (Max-Min) / Max.
The work is interesting and deserves to be published in a magazine.
Reviewer 3 Report
Comments and Suggestions for Authors
I have reviewed the manuscript titled "Allometric growth of Pinus yunnanensis after decapitation under different shading levels". In my opinion, the authors should address the following issues to improve the manuscript:
There are many typos and spelling errors in the manuscript, such as lines 48, 196, 261, ....
What specific scientific hypothesis or knowledge gap does the study test regarding allometry under shading? Please indicate this in the introduction section.
The sample size and replication structure are confusing. The text mentioned 768 seedlings, but only “4 technical × 3 biological replicates” were used. This must be clarified in the method section.
Table 1 (experimental layout) appeared redundant; it is unnecessary.
The discussion repeated background information rather than interpreting results. Many sentences or points were repeated almost verbatim across the Introduction, Discussion, and Conclusion.
Unify the citation style for the reference list.
Round 2
Reviewer 3 Report
Comments and Suggestions for Authors
Thank you for providing the revision. The authors responded positively to all comments and improved the quality of the manuscript. Therefore, the present version can be accepted for publication.
